# Fractionation of Winemaking Grape Stalks by Subcritical Water Extraction to Obtain Added-Value Products

**DOI:** 10.3390/foods13223566

**Published:** 2024-11-07

**Authors:** Irene Maté, Maria Vargas, Lorena Atarés, Amparo Chiralt

**Affiliations:** Instituto Universitario de Ingeniería de Alimentos-FoodUPV, Universitat Politècnica de València, 46022 Valencia, Spain; irmagar1@upv.edu.es (I.M.); loathue@tal.upv.es (L.A.); dchiralt@tal.upv.es (A.C.)

**Keywords:** hydrothermal treatment, integral valorisation, lignocellulosic waste, hydrogen peroxide bleaching, phenolic compounds, antioxidant, cellulose fibres

## Abstract

Grape stalks (GSs) from winemaking were submitted to a green process to valorise its lignocellulosic biomass that applied subcritical water extraction (SWE) at 170 °C and 180 °C to obtain active extracts and cellulose-enriched fractions. The sum of the total phenolic content of the soluble extract and the solid residue fractions from the SWE exceeded that of the GS, which suggests the generation of compounds with antioxidant properties through SWE. All SWE fractions showed high antioxidant power. The increased temperature promoted the extraction of polyphenolic compounds, enhancing the antioxidant power of both extracts and solid residues. These solid residue fractions were bleached with alkaline hydrogen peroxide solutions (4 and 8% *v*/*v*) to purify cellulose. After two bleaching cycles, no notable delignification progress was observed, as the bleaching yield or whiteness index did not significantly change in the further cycles. The first bleaching cycle led to a significant reduction in the lignin content at both SWE temperatures. The cellulose purity was higher in the samples obtained at 170 °C and bleached with 4% alkaline hydrogen peroxide. SWE at 180 °C led to greater cellulose oxidation during the bleaching step regardless of the hydrogen peroxide concentration.

## 1. Introduction

Grapes are one of the most abundant fruit crops in the world. In 2022, the world’s vineyards of approximately 7.3 million ha produced a total of 80.1 million tons of fresh grapes. Of the total volume available, 37.3 million tons were destined for pressing (34.1 million tons to produce wine and 3.2 million tons for the production of musts and juices) [1]. Grape stalks constitute a significant by-product of wine production, representing up to 7% of the raw material [2,3,4]. The production of 100 L of white or red wine yields approximately 4 kg of grape stalks [2]. Even though grape stalks are not hazardous waste, their high content of organic matter and the seasonal production of grapes represent an environmental problem. The discharge of grape stems into the soil inhibits its germinative properties [3] and may affect groundwater’s chemical and biological oxygen demand [2,5]. Grape skins can be used for animal feed, whereas grape stalks are mainly utilised as fertilisers [2]. However, the chemical composition of grape stalks offers interesting prospects for their valorisation by obtaining phenolic compounds, cellulose, or lignin in the context of the circular economy by using sustainable processes with low environmental impact.

The grape stalk is mainly composed of cellulose (12–36%), hemicellulose (14–26%), and lignin (23–34%) [5,6,7,8], as well as other non-structural components extractable in water or organic solvents (tannins, other polyphenols, etc.), proteins, waxes, and ashes. Compositional discrepancies in the literature may arise from using different analytical procedures, varying definitions of biomass fractions, and differing origins and types of grape stalks. The valorisation of lignocellulosic materials such as grape stalks is linked to the potential applications of some components, mainly the cellulosic fraction and the extractable phase with antioxidant/antimicrobial properties. The demand for cellulose fibres has been increasing recently and is expected to grow [3,9]. The environmental impact of cellulose extraction from wood pulp, namely forest devastation and its consequential contribution to global warming, have driven research efforts toward utilising more environmentally friendly alternative cellulose sources, such as lignocellulosic agricultural residues [9]. Exploring grape stalk as a cellulose source has been carried out by other authors [3,5,10] but using a non-environmentally friendly process with a great amount of chemicals for pulping and bleaching treatments. Araújo et al. [3] evaluated cellulose extraction from grape stalks using sequenced acid hydrolysis, alkaline hydrolysis, and a single 24 h bleaching step with alkaline hydrogen peroxide, performed on samples of varying particle sizes. Delignification, indirectly evaluated with colour measurements, was more efficient in small particles with low cellulose yields. Other fractionation processes have been tested for the valorisation of grape stalks, aiming for the release of fermentable sugars. In this regard, Atatoprak et al. [10] used an alkaline treatment for delignification, whereas Ping et al. [11] found that dilute sulfuric acid and ethanol organosolv pretreatments resulted in low degrees of delignification due to the high content of lignin and tannins of the feedstock.

Grape stalks also contain high proportions of non-structural functional compounds that have demonstrated favourable effects on human health [12]. These include polyphenols such as tannins, flavonols, and stilbenes [4]. Antioxidants are naturally present in many plant-based food products, such as nuts, oilseeds, vegetables, and fruits. Using low-cost industrial wastes instead of fresh raw materials for phenolic compound extraction could reduce production costs and increase the margin profit of these high-added-value products [12]. As well as other winemaking industry residues, grape stalks are considered a promising rich source for these functional compounds. Grape stalk extracts find, therefore, potential applications in the pharmacy and food industries, such as the shelf-life extension of fatty products, the inhibition of food pathogens, or the formulation of active food packaging [4]. Solvent extraction is the most frequently used procedure for antioxidant recovery [12]. Even though these processes may achieve efficient antioxidant extraction, using hazardous chemicals entails recovery costs and the environmental risks linked to their disposal. In contrast, subcritical water extraction, SWE (also known as hydrothermal treatment or autohydrolysis), using water as a solvent, accomplishes acidic hydrolysis without the use of any external acid because of hydrogen ions produced by the water autoionisation. This process results in the fractionation of lignocellulosic materials by using only hot compressed water, enabling the simultaneous removal of antioxidant water-soluble extractives, hemicelluloses, and soluble lignin. In this way, cellulose and Klason lignin quantitatively remain in the extraction solid residue, constituting a cellulose-enriched fraction that can be more easily purified by a delignification step. Difonzo et al. [13] reported on the potential of grape stalks as a source of antioxidant compounds extracted by this process. Freitas et al. [14] obtained antioxidant and antibacterial extracts from grape stalks (red and white varieties) by using SWE at 160 °C and 180 °C, which allows for active films for food packaging to be obtained. Amendola et al. [6] found that SWE efficiently extracted solid residues from grape stalks enriched in cellulose and lignin with lowered proportions of ashes and hemicellulose. Similar results were reported by Senila et al. [15], who additionally applied a subsequent delignification step with sodium chlorite in an acid medium to improve cellulose purification.

Cellulose pulping and delignification are usually carried out through different chemical treatments of lignocellulosic biomass procedures, such as Kraft pulping, alkaline or acid hydrolysis, ionic liquids, and oxidant delignification treatments such as acidified sodium chlorite [15,16]. All of these implied the use of great amounts of chemicals and water, making the process sustainable difficult. Therefore, reducing chemicals is essential to designing more environmentally friendly and lower-cost processes to purify cellulose. In this sense, the present study proposes the use of hydrogen peroxide in the bleaching process, which allows for operation under mild conditions without forming chlorine-toxic by-products that are recalcitrant for biodegradation. In alkaline media, hydrogen peroxide dissociates to yield hydroperoxyl anions (HOO^−^) and superoxide ions (O_2_^−^), which cleave side chains in lignin and oxidise chromophores. These potent antioxidants produce delignification, yielding low molecular weight water-soluble products, but cellulose depolymerisation could also occur as an undesirable side-effect in achieving efficient lignocellulose fractionation [15,17,18]. In previous studies, hydrogen peroxide treatment in an alkaline medium has been successfully used to obtain cellulose fibres from rice straw [14], almond skins [19] and almond shells [20], *Posidonia oceanica* waste [21] or vineyard pruning residues [17]. However, the combined effect of subcritical water treatment and hydrogen peroxide delignification for obtaining cellulose-enriched fractions from winemaking grape stalks has not been evaluated yet.

This study aimed to evaluate the potential of using subcritical water extraction and hydrogen peroxide alkaline delignification to fractionate winemaking grape stalks, producing phenol-rich aqueous extracts and cellulosic fractions, both valuable in developing food packing materials. Both extracts and solid residues were characterised in terms of their phenolic richness and antioxidant activity. The structural components (cellulose, hemicellulose, and lignin) were analysed in the solid residues of the extraction and at the different steps of the bleaching process to assess the efficacy of the fractionation treatments.

## 2. Materials and Methods

### 2.1. Materials

Grape stalk (GS) residues from red grape variety Bobal, origin Utiel-Requena (Valencia, Spain), were supplied by a local wine producer after the winemaking process. Folin–Ciocalteu reagent (2 N), gallic acid, methanol (>99.9 purity), 2,2-Diphenyl-1-picrylhydrazyl (DPPH), sodium hydroxide, glucose, and arabinose were purchased from Sigma-Aldrich (St. Louis, MO, USA). Ethanol (98%), hydrogen peroxide (H_2_O_2_, 30%), sulphuric acid (H_2_SO_4_, 98%), and sodium carbonate (Na_2_CO_3_, 99.5%) were obtained from Panreac Quimica S.L.U (Castellar del Vallés, Barcelona, Spain). Phosphorous pentoxide (P_2_O_5_, 98.2%) was obtained from VWR Chemicals (Leuven, Belgium). D(+)-Xylose was supplied by Merck KGaA (Darmstadt, Germany).

### 2.2. Subcritical Water Extraction of Grape Stalks

The grape stalk raw material was dried at 105 °C for 24 h and ground with a mill using a 1 mm diameter mesh (Model Zyklon SM 300 stainless steel, Retsch, Haan, Germany). The milled GS was sieved with a 0.55 mm mesh and stored at 0% RH until further use.

The milled and sieved GS was subjected to subcritical water extraction (SWE) in a pressure reactor (Model 1-TAP-CE, 5 L capacity, Amar Equipment PVT. Ltd., Mumbai, India) at a solids/water mass ratio of 1:10. The extraction was carried out for 30 min, and two temperatures were tested, 170 °C and 180 °C. These temperatures were selected based on previous studies that report a higher hydrolysis of hemicellulose near 180 °C [14]. Subsequently, the extraction residues were left to decant for 3 h to separate the liquid extract and the solid residue. The liquid extracts were filtered with a qualitative filter (Filter-lab S.A., Barcelona, Spain, pore size < 0.5 mm), frozen at −40 °C and subsequently freeze-dried (Telstar, model Lyoquest-55; Barcelona, Spain) at −45 °C, 6 mbar, and 72 h. The lyophilised extracts (E-170 and E-180) were stored at −20 °C until further use.

The solid extracts obtained at either temperature (R-170 and R-180) were washed with distilled water, filtered with a qualitative filter (pore size < 0.5 mm, Filter-lab S.A., Barcelona, Spain,), and dried at 50 °C for 24 h. The dried mass of the extracts and solid residues were used to analyse mass balance in the SWE process and quantify the yield of each fraction.

### 2.3. Characterisation of Phenolic Compounds and Antioxidant Activity of SWE Fractions

Total phenolic content (TPC) and antioxidant activity with the DPPH assay were determined in the raw material (GS), in the soluble extracts (E-170 and E-180), and in the insoluble fractions (R-170 and R-180) obtained from SWE. For raw material and insoluble residues (GS, R-170, and R-180), an ultrasound extraction of phenols in methanol-water mixtures (40:60 *v*/*v*) at 1:10 S/L mass ratio and 30 °C was performed before the quantification test. To this aim, a high-intensity ultrasonic homogeniser was used (Vibra CellTM VCX750, Sonics & Material, Inc., Newtown, CT, USA), applying a frequency of 20 kHz, a power of 750 W and a sonication amplitude of 40% for 30 min.

TPC was determined using the Folin–Ciocalteu method in triplicate, according to the procedure described by Freitas et al. [22]. The assay consisted of mixing 0.5 mL of extract with 6 mL of distilled water and 0.50 mL of Folin reagent. After one minute, 1.5 mL of a 20% Na_2_CO_3_ solution was added and made up to 10 mL with distilled water. It was then shaken and kept for 2 h in the dark at room temperature. After that time, absorbance was measured at 750 nm using a UV–Vis spectrophotometer (model Evolution 201, Thermo Scientific, Waltham, MA, USA). Finally, TPC was determined using a gallic acid calibration curve (R^2^ = 0.9991), and the results were expressed as g gallic acid equivalents (g GAE) per 100 g of sample solids. The results were also expressed as g GAE per 100 g of grape stalk to evaluate phenolic balance in the fractions.

The ability of GS and SWE fractions to scavenge DPPH free radicals was determined using the 2,2-Diphenyl-1-pikryl-hydroxyl (DPPH) method, with some modifications [23]. Each sample extract was mixed with a methanolic solution of DPPH at different ratios. The reaction was left to complete in the dark at room temperature, and the absorbance was measured at 515 nm. The initial and final DPPH concentration was obtained from a calibration curve fitted by linear regression (R^2^ = 0.9992). The EC50 was determined, which is defined as the ratio of sample to DPPH that is required to reduce the initial concentration of DPPH to 50% once the stability of the reaction has been reached. Thus, the lower the EC50 values, the greater the antioxidant activity of the tested sample. The results were expressed in mg sample per mg DPPH and mg GS per mg DPPH.

### 2.4. Cellulose Purification from the Insoluble SWE Fractions

Aiming to purify cellulose fibres from the insoluble SWE fractions (extraction residues), the solid extracts were bleached with hydrogen peroxide solutions at a 1:30 S/L mass ratio at pH 12. Two different hydrogen peroxide concentrations were tested, namely 4 and 8% *v*/*v*. Four consecutive 1 h bleaching cycles at 40 °C were performed, and after filtering and washing with abundant distilled water, the samples obtained in each cycle were air-dried at room temperature for three days, and the mass yield was determined in each step. The bleached residues were coded as BR-170-4, BR-170-8, BR-180-4, and BR-180-8.

The colour coordinates L* (lightness), a* (red-green), and b* (yellow-blue) of each bleached fraction were measured with a CM-3600d spectrocolorimeter (Minolta Co., Tokyo, Japan). The effectiveness of the bleaching treatment was evaluated through the mass yield and the whiteness index (WI) calculated with Equation (1):(1)$$WI=100–\sqrt{{\left(100-L*\right)}^{2}+{a*}^{2}+{b*}^{2}}$$

Additionally, the morpho-geometric characteristics of the fibres after different process steps were qualitatively evaluated using a field emission scanning electron microscope (ULTRATM 55, Zeiss, Oxford Instruments, UK). The conditioned samples (P_2_O_5_ at 25 ± 2 °C for one week) were covered with a gold layer (EM MED020 sputter coater, Leica Biosystems, Barcelona, Spain), and the images were taken at 2.0 kV acceleration voltage.

#### 2.4.1. Analysis of Structural Components in the Lignocellulosic Fractions

The content of cellulose, hemicellulose and lignin of GS, the extraction solid residues (R-170 and R-180), and the bleached materials (BR-170-4, BR-170-8, BR-180-4, and BR-180-8) after different bleaching cycle were determined. The extractive contents in ethanol and water were previously determined in duplicate in non-bleached samples according to the NREL standard procedure (NREL/TP-510-42619-2008) [24]. After eliminating the non-structural components, structural carbohydrates (cellulose and hemicellulose) and lignin were quantified according to the NREL standard procedure (NREL/TP-510-42618-2008) [25]. This method involves acid hydrolysis with sulphuric acid in two steps and a filtration step to separate the soluble and insoluble hydrolysed fractions. The insoluble hydrolysed fraction was washed with abundant water and dehydrated at 105 °C for two days to quantify the acid-insoluble fraction (AIF) gravimetrically. The insoluble lignin was determined from the AIF by subtracting the ash content of AIF determined by incineration at 550 °C. The free sugars resulting from the hydrolysis of structural carbohydrates (glucose, xylose, and arabinose) were determined in the soluble hydrolysed fraction.

Free sugars (glucose, xylose, and arabinose) were quantified by HPLC (Agilent Technologies, model 1120 Compact LC, Frankfurt, Germany). A HILIC Luna Omega Sugars HILIC column (150 × 4.6 mm, three μm) and an evaporative light scattering detector (ELSD Agilent Technologies 1200 Series, Waldbronn, Germany) were used. The mobile phase consisted of water: acetonitrile (25:75) in isocratic mode at a 0.8 mL min-1 flow rate. The detector conditions were 40 °C, 3.0 bar N_2_ pressure, and a gain of 3. The software used for data acquisition was ChemStation (version LTS 01.11, Agilent Technologies, Waldbronn, Germany). Results were analysed using the Origin program (version OriginPro 2021, OriginLab Corporation, Northampton, MA, USA) by applying the Gaussian model for peak area determination. The hemicellulose content was calculated from the sum of the xylose and arabinose concentrations, and the cellulose content was obtained from the glucose concentration by applying the corresponding correction factors related to the water molecular weight, as described by Sluiter et al. [24]. The cellulose content would be overestimated because glucose units are also present in the hemicellulose chains, whereas hemicellulose content would be underestimated.

The ash contents in the different samples were quantified as the mass percentage of residue remaining after incineration at 550 °C for 24 h (NREL/TP-510-42622-2008) [25].

#### 2.4.2. Thermal Analyses (TGA) of the Lignocellulosic Fractions

The thermal behaviour of the different samples was analysed with a thermogravimetric analyser under a nitrogen atmosphere (TGA 1 Stare System analyser, Mettler-Toledo, Greifensee, Switzerland). Before the tests, the samples were placed in a desiccator with P_2_O_5_ for two weeks to eliminate moisture. The TGA test consisted of heating from 25 °C to 900 °C at 10 °C min^−1^ under a constant nitrogen flow (10 mL). All samples were run in duplicate. The results were evaluated using STARe software (version V12.00a, Mettler-Toledo, Inc., Greifensee, Switzerland).

#### 2.4.3. FTIR Analyses of the Lignocellulosic Fractions

An FTIR spectrophotometer (Vertex 80, Bruker AXS GmbH, Karlsruhe, Germany) was used to evaluate the vibrational patterns of the functional groups present in the samples (GS, R-180, R-170, and bleached samples). FTIR spectra were obtained at a 6 cm^−1^ resolution, in the wavelength range 4000–650 cm^−1^, and 128 scans were performed for each spectrum. The measurements were performed in duplicate.

#### 2.4.4. X-Ray Diffraction Analysis (XRD) of the Lignocellulosic Fractions

An X-ray diffractometer (AXS/D8 Advance, Bruker, Karlsruhe, Germany) was used to analyse the X-ray diffraction pattern spectra of the GS and the different cellulosic fractions (R-170, R-180, BR-170-4-4, BR-170-8-4, BR-180-4-4, and BR-180-8-4). The analyses were performed using Kα-Cu radiation (λ: 1.542 Å), 40 kV, 40 mA, a step size of 2.0° per minute, and a scan angle of 2θ between 5° and 40°. Once conditioned at 53% RH and 25 °C, the samples were appropriately compacted on the sample holder. The data were obtained using XRD Commander software (version 8.61, Bruker AXS/D8 Advance, GmbH, Karlsruhe, Germany) and processed with DIFFRAC.EVA and DRXWin (Windows, version 2.3) software. The crystallinity index (CI%) was determined from the spectra using Equation (2) [26], where the maximum intensity of 200 lattice diffraction (I_200_, crystalline peak) and the diffraction intensity at 2θ = 18° (I_2θ_ 18°, valley of the amorphous phase) are related.
(2)$$CI\left(\%\right)=\frac{\left(I_{200}-I_{2\theta \;18{}^{\circ}}\right)}{I_{200}}100$$

### 2.5. Statistical Analysis

The experimental data were submitted to analyses of variance (ANOVA) with a 95% confidence level using Statgraphics version 19-X64. Comparisons were performed using 95% Fisher’s LSD intervals.

## 3. Results and Discussion

### 3.1. Yields of the SWE Process and Composition of the Grape Stalk and Extraction Residues

Figure 1 shows the flow chart of the process applied to revalorise grape stalks, where the values of mass yields (Y: dry out-flow mass to dry in-flow mass) of the different fractions obtained in the SWE process steps at both 170 °C and 180 °C, are specified.

The extraction step with subcritical water at 170 °C and 180 °C showed a solid extraction yield of 32% and 29%, respectively, in the range of those reported by Amendola et al. [6]. This yield indicated a high water solubilisation of the GS components under the conditions applied. The dry, solid, insoluble residues yield represented 54% and 58% for R-170 and R-180, respectively, in the range of those found in previous studies. In this sense, Senila et al. [15] performed the autohydrolysis of vine-shoots waste of different grape varieties at 165 °C and 180 °C and reported solid yields ranging between 55% and 70%. In the present study, no significant differences were observed between the yields in the lignocellulosic insoluble residue at 170 °C and 180 °C. Still, the R-170 sample could have more cellulose since more extraction of non-cellulosic components occurred at 170 °C.

The sums of the yields of soluble fractions (E) and the insoluble fractions (R) were 86% and 87% at 170 °C and 180 °C, respectively, which corresponds to a mass loss of 13–14%. This could be attributed to the experimental variability and the partial degradation of the organic matter caused by hydrolysis (e.g., loss of formic or acetic acids resulting from hydrolysis of methoxy or ethoxy groups in some molecules) at these temperatures. In fact, a slight pressure increase was detected in the reactor during the extraction processes (up to 1.5 bars above the initial pressure), which can be attributed to the volatile formation in the closed reactor.

The contents of extractives and structural components of the GS and the SWE solid, insoluble residues (R-170 and R-180) obtained at 170 °C and 180 °C are shown in Table 1.

In the GS samples, more than 50% of GS solids were water-extractable in Soxhlet, whereas no ethanol-soluble compounds were quantitatively extracted. These water-soluble compounds must be mainly tannins and free sugars, as reported by other authors for other GS grape cultivars [1,2,5]. In the SWE extraction residues (R-170 and R-180), the extractive amount was lower, due to the previous extraction of some of these constituents during the SWE step. SWE at 170 and 180 °C resulted in the removal of an important fraction of hemicellulose from the grape stalk, as has been previously described for different lignocellulosic materials [19,27,28]. A high hydrolysis degree of the hemicellulose at 160 and 180 °C was also reported in GS by Freitas et al. [14]. The lignin and cellulose contents significantly increased in R-170 and R-180 samples due to the significant removal of water extractives and hemicellulose during the SWE step. In contrast, the ash content was significantly reduced without a significant effect of temperature (*p* > 0.05).

The cellulose content in the GS was approximately 22%, in the range previously reported for the cellulose content GS (12 to 36%, depending on the grape variety and origin) [1,2,6,7,29]. After the SWE step, the grape stalk residues exhibited an increased cellulosic content, thus reflecting the cellulose enrichment provoked by the process. This has also been found in other lignocellulosic materials such as rice straw [14,30], defatted almond skin [19] or the waste of the Posidonia oceanica leaves [21]. When the cellulose content of R-170 and R-180 samples was referred to per mass unit of GS, taking the process yield into account, it can be noted that only 14–17 g cellulose/g GS was obtained. This could suggest a cellulose loss during the SWE process by the autohydrolysis process. Nevertheless, the difference could also be attributed to the overestimated value of the cellulose content of the raw GS, where more hemicellulose is present and more hemicellulosic glucose would be released during the hydrolysis step with sulphuric acid.

### 3.2. Total Phenol Content and Antioxidant Properties of the GS and SWE Fractions

The total phenolic content (TPC) and antioxidant activity (EC_50_ values) of the GS and the different SWE fractions obtained at 170 °C and 180 °C are reported in Table 2. These results are shown per mass unit of the sample solids and per mass unit of GS, which enables the analysis of the phenolic mass balance in the corresponding fractions.

Subcritical water extraction allowed for obtaining phenolic-rich extracts with more than 70% of the TPC of GS, whereas a part of these compounds also remained in the solid fraction. The highest temperature provided the extract E-180 with greater phenol concentration, according to a greater hydrolysis degree of the biomass and release of bonded phenols. At a determined temperature, the sum of the TPC of both E and R samples from SWE exceeded the TPC value of the initial GS material, mainly at 180 °C. This suggests the generation of some antioxidant compounds that are reactive with the Folin reactant at high extraction temperatures. This could be attributed to the release of lignin-derived polyphenol antioxidants, as has been previously observed by other authors [31] or to the formation of Maillard and caramelisation compounds at high temperatures, as reported by different authors due to the reaction of free sugars [28]. The neo-formed brown compounds, such as 5-hydroxymethylfurfural (HMF), exhibit antioxidant properties [28] and can be sensitive to the Folin reactant. This implies the enhancement of the antioxidant capacity of the biomass fractions, but the potential toxicity of the neo-formed compounds must be analysed.

The TPC found for the extracts was significantly higher than others previously reported for the soluble fraction of grape stalk obtained by SWE [6,13,14]. They were also higher than TPCs obtained from grape stalks using solvent extraction procedures for antioxidant compound recovery. In this sense, Spigno et al. [12] extracted polyphenols from the grape stalk using ethanol and ethylacetate as solvents at varying temperatures and time, reporting less than 0.4 g GAE per 100 g of GS, which is significantly lower than the values obtained in the present study.

The values of EC_50_ supported the high antioxidant power of all obtained GS fractions in the range of pure, potent antioxidants, like ascorbic acid or tocopherol (0.12 and 0.26 mg compound.mg^−1^ DPPH, respectively [23]. Freitas et al. [32] found 0.6 and 1.1 mg extract mg^−1^ DPPH for the EC_50_ values of SWE extracts from GSs of red and white grape varieties, respectively, which is in the range of the obtained values. This demonstrates the great antioxidant power of the grape stalk extracts obtained by SWE. The variability in the phenolic extraction efficiency can be attributed to different factors, such as the biomass variety and origin and the sample pretreatments (drying and milling), which may affect the solvent penetration and the polyphenol recovery, depending on the temperature applied in the SWE [33,34]. Cheng et al. [34] reviewed the optimal temperature/time conditions for extracting polyphenols from different plant matrices and found that the optimal values ranged from 120 to 225 °C. Increased temperature within a determined range seemed to promote the extraction of polyphenolic compounds from the grape residue feedstock. Gullón et al. [35] found the same positive effect of temperature on TPC and DPPH radical scavenging capacity of SWE extracts of dry vine shoots between 180 °C and 215 °C.

The phenolic richness of the SWE extraction residues (R-170 and R-180) is also remarkable since these retained approximately 30–40% of the total phenols of GS. The corresponding EC_50_ values also reveal that these residues have as strong antioxidant power as the extracts. The EC_50_ values referred to per mass unit of GS showed that their antioxidant capacity is approximately 10 times higher than that of the raw GS. This fact suggests that even after the extraction process, the solid SWE residues could represent an attractive source of antioxidant compounds potentially applicable in various fields. For instance, these residues could be used as fillers of biodegradable polymeric matrices to obtain lower-cost, reinforced, and potentially active food packaging materials.

### 3.3. Obtaining Cellulosic Fibres from SWE Insoluble Residues

The cellulose-enriched extraction residues (R-170 and R-180) obtained from SWE at 170 °C and 180 °C were submitted to four successive bleaching steps with alkalinised H_2_O_2_ to promote cellulose delignification. Figure 2a shows the bleaching yield (BY, g of bleached sample per 100 g of SWE residue) after one to four bleaching cycles together with the corresponding whiteness index (WI) as indicators of the delignification progress. Figure 2b shows the appearance of the samples after the successive process steps.

Bleaching treatments allowed for the elimination of both coloured and non-coloured compounds of the cellulosic residue that are sensitive to the action of oxidants, while their oxidised forms are solubilised in the bleaching medium, thus reducing the mass of the solid residue. Bleaching yields after the fourth cycle ranged between 25% and 40%, which implies that a great part of the components of the SWE residues were susceptible to alkaline oxidation and, therefore, solubilised in the liquid medium. Bleaching yields were higher for the R-170 samples than for R-180, which indicates a more significant elimination of non-cellulosic components in the residue obtained at 170 °C. The concentration of hydrogen peroxide did not significantly affect bleaching yield in R-180 samples, whereas the 8% concentration of H_2_O_2_ was more effective in R-170 samples, but only during the first bleaching cycle. In this sense, the minute change in the blanching yield after the second cycle is remarkable, which suggests that no further compound elimination occurred after this cycle.

Concerning the whiteness index (WI), which allows for controlling the removal of coloured compounds, relatively low WI values were observed for the R-170 and R-180 samples without significant differences between them (Figure 2a,b). The severe conditions used for SWE promote caramelisation or Maillard reactions in the GS substrate, as previously observed by other authors in terms of rice straw [14], which resulted in deeply coloured extraction residues. The development of WI throughout the bleaching cycles revealed very similar behaviour of both R-170 and R-180 samples with 4 or 8% H_2_O_2_. Only slight differences could be appreciated after the first cycle, where treatments with 8% H_2_O_2_ provided the samples with a slightly higher WI value. It is remarkable that even after four bleaching cycles, the samples did not show WI values greater than 60, which indicates the recalcitrance of GS-coloured components to the peroxide treatment. This can be attributed to the high tannin content of GS, which makes delignification difficult. Senila et al. [15] reported that the high proportion of tannins in the vine-shoot waste, structurally associated with the lignin fraction, through the formation of alkali–aryl condensed structures, makes the separation difficult. The persistence of coloured compounds in bleached cellulosic material obtained from grape stalks was also observed in previous studies [3].

Structural components were also analysed in the samples R-170 and R-180 after one and two bleaching cycles, with 4% and 8% hydrogen peroxide solutions, since no progress in the cellulose purification after two cycles can be deduced from the bleaching yield development (Figure 2a). Table 3 shows the content of cellulose, lignin, and ashes of the different lignocellulosic residues, referred per mass unit of GS to evaluate the mass balances better, and the global yield of the fraction submitted to the corresponding process steps. Hemicellulose was not reflected in Table 3 since no quantitative amounts of xylose and arabinose could be detected in the bleached samples. Therefore, the small hemicellulose amount in the extraction residues R-170 and R-180 (Table 2) was practically eliminated after the first alkaline bleaching cycle. Likewise, approximately 65–75% of lignin from GS was removed during the two bleaching cycles without significant effect on the hydrogen peroxide concentration. After bleaching, the R-180 sample had slightly lower lignin content than R-170, which suggests a better aptitude for bleaching of the sample extracted at the highest temperature, as previously observed for other lignocellulosic residues submitted to a previous SWE step [14,20]. The combined treatments provoked a remarkable reduction of ashes in the cellulosic fractions due to their progressive lixiviation in the liquid phases promoted by the weakening of the matrix and the increased exposure area during the different treatments. Figure 3 shows the contents (g/100 g sample) of cellulose and lignin reached after the two bleaching cycles in comparison to those of the non-bleached residues (R-170 and R-180).

The results indicated that alkaline bleaching with hydrogen peroxide was not effective in removing the total lignin content of the SWE residues (R-170 and R180). However, a significant reduction of lignin content was observed in both materials, even at the first bleaching cycle, without a remarkable effect of hydrogen peroxide concentration. The final lignin content in the bleached samples (more than 20%wt.) was higher than those previously reported for other lignocellulosic matrices, such as rice straw [14] or almond shells [20] by applying a similar combined process (SWE plus alkaline bleaching with hydrogen peroxide). This can be mainly attributed to the high lignin content of GS and the particular structural association of lignin with other macromolecular components of grape stalks [2]. Ping et al. [11] also achieved low degrees of delignification of grape stalks using dilute sulphuric acid and ethanol organosolv treatments. These authors also stated that the valorisation of grape stalks as a cellulose source is challenging due to their high content of lignin and tannins.

Concerning the cellulose yield obtained in the different treatments, expressed as the mass of cellulose recovered with respect to that of raw GS, the overestimated concentration of this component in GS samples by the applied quantification method should be considered. A more realistic cellulose content would be determined in the SWE residues where hemicellulose is present in a very small proportion. Thus, considering 14 g/100 g of GS (as determined in the R-180 sample with the lowest hemicellulose content), a practically total recovery of cellulose from the R-170 samples was achieved after the two bleaching cycles. In contrast, approximately 25% of the GS cellulose was lost during the bleaching treatments of R-180 samples. This loss could be attributed to the cellulose oxidation by the oxygen radicals through the formation of alpha hydroxyalkyl radicals, giving rise to chain scission and carbonyl compounds [36]. The greater progress of this cellulose degradation in the R-180 sample could be attributed to the greater exposure area of this sample, where a higher extraction ratio was promoted by SWE, which facilitates the access of radicals to the cellulose points.

### 3.4. Characterisation of the Cellulose Fibres

The properties of the different lignocellulosic fractions were studied in terms of their morphological characteristics, FTIR analysis, X-ray diffraction patterns, and thermal degradation. Figure 4 shows representative FESEM images of the two-bleaching-cycle fibres from R-170 and the R-180 samples in comparison with the non-bleached fibres (R-180 and GS samples).

Untreated GS fibres showed uneven surfaces, where the fibrils are bound together by hemicellulose and lignin, which is in line with previous observations [37]. The SWE insoluble residues showed smoother surfaces than untreated GS, which agrees with the removal of different surface compounds during the SWE treatment. The structural changes caused by SWE are linked to the compositional modifications caused by the process, such as the quantitative extraction of hemicellulose and partial removal of lignin, causing relevant morphological changes in the fibre structure. Bleached samples still showed an increase in the fibre surface smoothness and some defibrillation without appreciable differences due to the different SWE temperatures and hydrogen peroxide concentrations. This morphological change must also be related to the progressive removal of the residual hemicellulose and lignin, giving rise to the typical morphology of more purified cellulose fibres previously described by other authors [37,38]. Thus, the bleached fibres achieved less cohesive entanglement fibrils in the bundles, with clear voids previously filled with other non-cellulosic constituents.

The FTIR spectra of the non-bleached (GS, R-170, and R-180) and bleached fibres are shown in Figure 5. The bands agreed with those reported in previous studies for GS materials [14,39]. A strong absorbance band due to OH^−^ stretching at 3600–3000 cm^−1^ was observed, which is associated with the presence of this moiety in cellulose, hemicellulose, and lignin. The two peaks at approximately 2900 cm^−1^ correspond to the asymmetric and symmetric vibration of C-H, respectively. A strong band at 1033 cm^−1^ is found due to C-O stretching and bending. Characteristic lignin peaks appear at 1606, 1509, and 1423 cm^−1^, assigned to aromatic skeletal vibrations of syringil and guayacil aromatic rings [8,40]. Likewise, the band at 1730 cm^−1^ is attributed at C=O in ester groups, present in phenolic and uronic acids, and aromatic compounds in lignin and tannins. The band corresponding to the β-linkage of cellulose appears at 898 cm^−1^ [41].

After SWE treatment, the carbonyl vibration band practically disappears from the FTIR spectra. In contrast, the vibration band of cellulosic β-linkage was enhanced, according to cellulose enrichment of the samples with the hemicellulose removal and other sugars and phenols. After bleaching (Figure 5), the typical vibration bands of lignin aromatic rings still had a notable intensity in the FTIR spectra, along with the remaining amount of these compounds in the fibres, as deduced from the compositional analysis (Figure 3).

In lignocellulosic natural fibres, cellulose shows a semicrystalline structure, unlike hemicellulose and lignin, which are amorphous and have no effect on total crystallinity [38]. The X-ray diffraction spectra of GS, the SWE-insoluble fractions (R-170 and R-180), and the bleached residues (BR) coming out from the second bleaching at either 4% or 8% hydrogen peroxide (BR-170-4, BR-170-8, BR-180-4, and BR180-8) are shown in Figure 6, where the diffraction peaks characteristic of the type I cellulose crystalline lattice at 2θ = 15°, 16°, 22°, and 34° have been indicated [42].

GS presented an X-ray pattern similar to previous studies, with two low-intensity peaks at around 15–16° and 22° [38,43] and a low crystallinity index. The extraction of non-cellulosic compounds during the SWE treatments only implied a slight increase in the crystallinity index of the material while promoting the intensity of the peaks at 15°. In contrast, the bleaching step had a noticeable effect on crystallinity. The progression of cellulose purification with the partial elimination of the amorphous fractions entailed a remarkable increase in the crystallinity index. At the same time, the diffraction peak at 22° became narrower and more defined, according to the typical cellulose X-ray diffraction pattern. No relevant differences in the crystallinity of the different bleached samples, with CI ranging between 54–58%, were found. These values are lower than those reported by Prozil et al. [2], who reported extremely high values (75.4%) compared with those found in purified cellulose from other biomass (e.g., 55–65% in wood). As the final lignin content was high in every bleached sample, a low crystallinity degree is expected in the fibres where the final cellulose content ranged between 60–70%, depending on the SWE residue and bleaching agent concentration. However, the BR-180 samples contained approximately 65% cellulose, regardless of the H_2_O_2_ concentration used, and the final cellulose contents of BR-170 samples were approximately 60 and 70%, respectively, for samples treated with H_2_O_2_ at 8 and 4%.

Figure 7 shows the thermal degradation behaviour of the non-bleached (GS, R-170, and R-180) samples and one-and-two-cycles-bleached fibres through the TGA curves and their corresponding derivative curves. The SWE treatment improved thermal stability due to the removal of low molecular compounds and a great part of hemicellulose, which degrade at lower temperatures. Thermal decomposition of GS occurred in several mass loss steps. The first event at approximately 100 °C corresponded to the loss of bonded water, as observed in previous studies [37,44]. The mass loss at approximately 300 °C was attributed to cellulose degradation, which occurred at a lower temperature than the usually reported for this pure polymer due to the lack of purity. The peak at approximately 200 °C in GS samples can be attributed to the degradation of hemicellulose, which is not observed quantitatively in the samples after SWE treatment due to hemicellulose removal. The peak degradation temperature of pure hemicellulose has been reported to be approximately 270 °C [39], but structural differences and the presence of other compounds can shift this temperature to lower values. After SWE, the removal of different low molecular compounds and hemicellulose was reflected in the displacement of the peak temperature, which was mainly attributed to the cellulose degradation towards a higher temperature, in accordance with the higher cellulose purity. The decomposition of pure lignin has been reported to occur gradually over a wide range of temperatures [39]. The observed peak between 400 °C and 600 °C in GS and the R-170 and R-180 samples can be assigned to the present lignin in the samples, whose content increased in the R-170 and R-180 residues due to the removal of other SWE extractables. SWE affected the composition of the lignin fraction, modifying its thermal stability and shifting the peak temperature towards lower temperatures. Condensed tannins in the GS could also contribute to this degradation step [38].

In the bleached samples, a sharper weight loss step associated with cellulose degradation was observed, which was in line with higher cellulose richness. However, the onset and peak temperature were shifted to lower values: 276–278 °C in the first bleaching cycle and 285 °C in the R-170 sample after the second cycle. Additionally, at T > 325 °C, a thermal resistant mass was observed in almost all the samples, which was supposed to be approximately 50 and 40% of the samples in the first and second bleaching cycles, respectively. The latter could be associated with partially degraded compounds by the oxidative treatment (including lignin), which become more thermally stable. In the two-cycles-bleached R-170 samples, this thermally stable mass was absent, and only a second degradation peak associated with the residual lignin could be observed. Differences in the peak degradation temperature of cellulose from R-170 and R-180 samples also point to the greater oxidation/depolymerisation of cellulose in samples SWE-treated at 180 °C than at 170 °C, as commented on above due to the greater cellulose exposure to the action of radicals.

## 4. Conclusions

Applying subcritical water extraction to grape stalk allows for obtaining phenolic-rich extracts and solid residues with high antioxidant capacity, which can be used in different industrial fields such as pharmaceutical, cosmetics, and food industries, including food packaging. The subcritical water extraction process partially removed lignin, and most of the hemicellulose from the solid fractions became enriched in cellulose. The increase in subcritical water extraction temperature promoted the extraction of polyphenolic compounds in the extracts and the neoformation of antioxidant brown compounds, enhancing the antioxidant power of both extracts and solid residues. Thus, subcritical water extraction is useful for designing sustainable grape stalk fractionation processes, obtaining soluble or dispersible products with high antioxidant activity.

Moreover, the recovery of cellulose from the insoluble fractions, obtained by subcritical water extraction at 170 or 180 °C, was possible by applying a green delignification process with alkaline hydrogen peroxide. The first bleaching cycle led to a maximum reduction in the lignin content of the insoluble fractions of 75% at both subcritical water extraction temperatures. After two bleaching cycles, the mass yield and the whiteness index did not significantly change, which indicated that no notable delignification progress occurred in the samples. However, the final lignin content in the bleached samples was still higher than 25%. The cellulose purity was higher in samples treated at 170 °C and bleached with 4% hydrogen peroxide than in those treated at 180 °C when greater cellulose oxidation during the bleaching step seems to be promoted regardless of the oxidant concentration. Therefore, despite the cellulose richness of grape stalk, its recovery and purification using sustainable processes need to be optimised in further studies.

## Figures and Tables

**Figure 1 foods-13-03566-f001:**
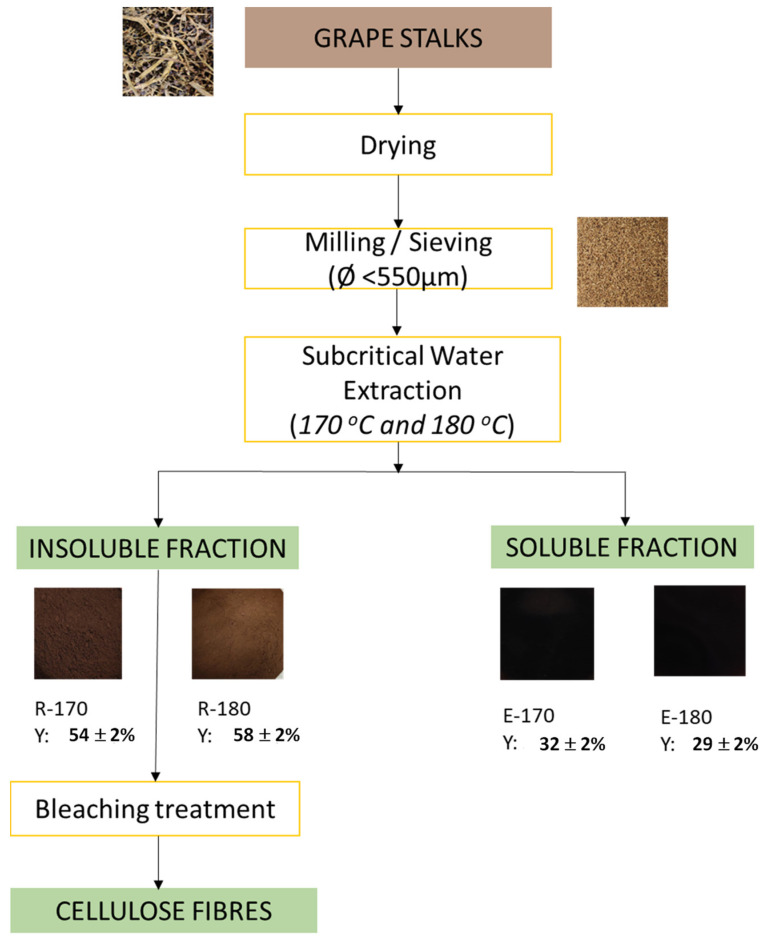
Flow chart of the process applied to valorise grape stalks. The Y-values represent the mass yield (dry out-flow mass with respect to the dry in-flow mass) of the different fractions obtained in the subcritical water extraction process at 170 °C and 180 °C. Mean values ± standard deviation. Insoluble residues (R-170, R-180) and extracts (E-170, E-180).

**Figure 2 foods-13-03566-f002:**
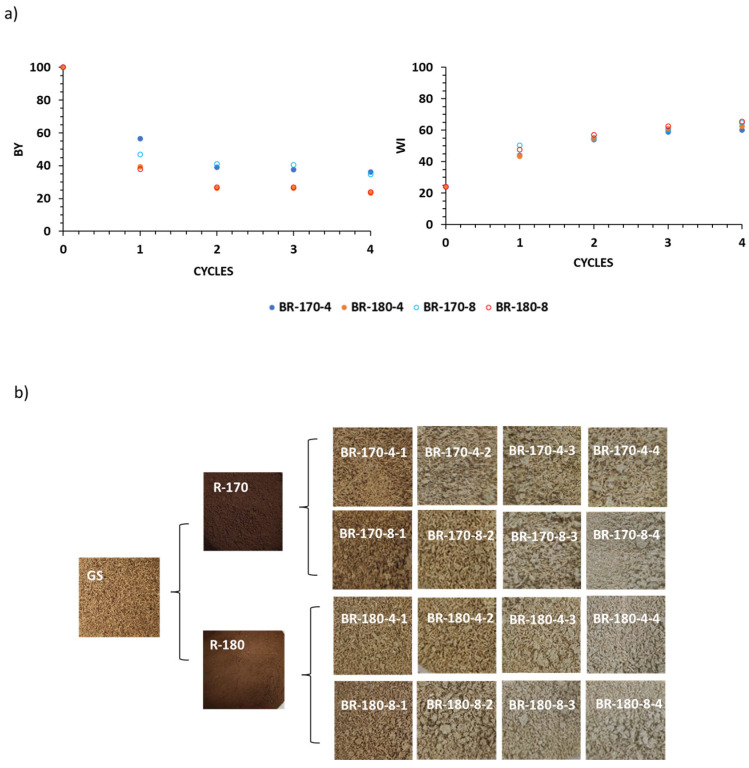
(**a**) Bleaching yield (BY) and whiteness index (WI) obtained with the different one-hour treatment cycles with alkaline hydrogen peroxide at 4 and 8% *v*/*v*. The LSD intervals are 2.22 for BY and 0.03 for WI. (**b**) Images of the raw material, subcritical water extraction (SWE) residues at 170 °C (R-170) and 180 °C (R-180), and bleached residues (BR) at the different steps of the bleaching process. Bleached residues are coded as the BR temperature of SWE concentration (% *v*/*v*) of alkaline hydrogen peroxide number of 1 h bleaching cycles.

**Figure 3 foods-13-03566-f003:**
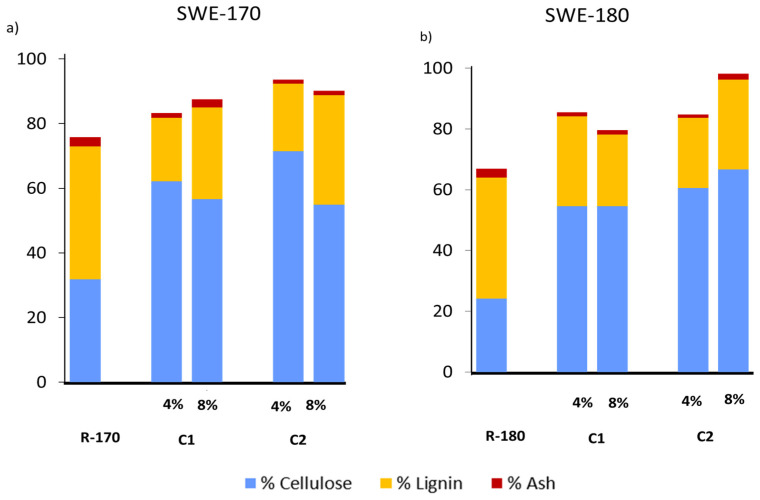
Cellulose, lignin, and ash content of residues after subcritical water extraction (SWE) at 170 °C (R-170) and 180 °C (R-180) and after two bleaching cycles (C1 and C2) with hydrogen peroxide solutions at 4 and 8% *v*/*v*. (**a**) Results for SWE at 170 °C (LSD intervals: 13.04 for cellulose, 8.10 for lignin, and 2.49 for ash). (**b**) Results for SWE at 180 °C (LSD intervals: 23.95 for cellulose, 11.63 for lignin, and 0.88 for ash).

**Figure 4 foods-13-03566-f004:**
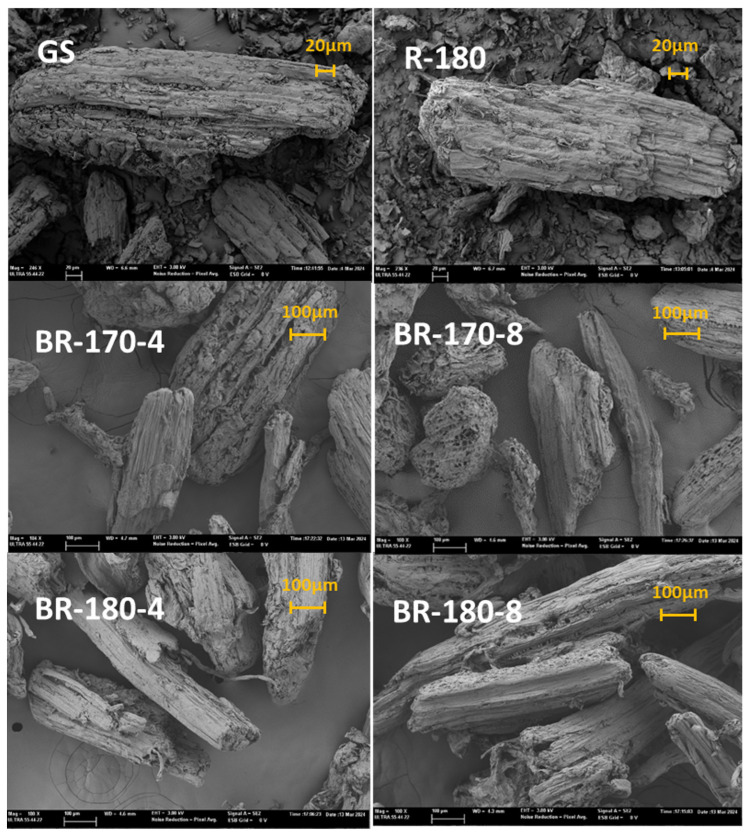
FESEM images of cellulose fibres after two bleaching cycles with 4% or 8% (*v*/*v*) alkaline hydrogen peroxide solutions (BR-170-4, BR-170-8, BR-18-4, and BR-180-8), compared with non-bleached fibres from subcritical water extraction at 180 °C (R-180) and grape stalk (GS) samples.

**Figure 5 foods-13-03566-f005:**
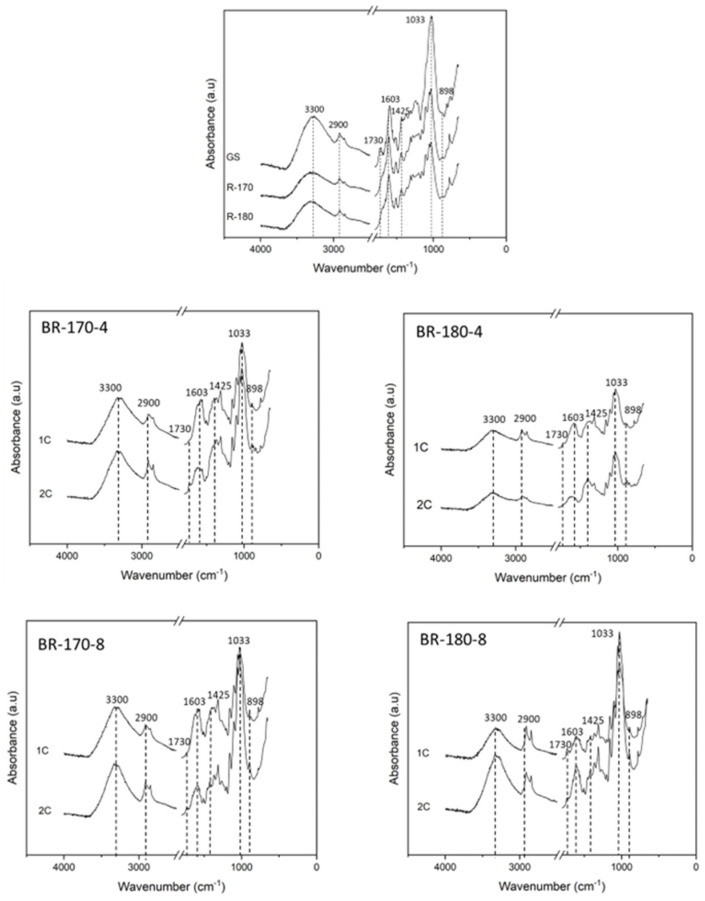
FTIR spectra of grape stalk (GS), non-bleached samples (R-170 and R-180) and samples bleached with 4 or 8% *v*/*v* alkaline hydrogen peroxide solutions (BR-170-4, BR-170-8, BR-180-4, and BR-180-8) after one and two bleaching cycles (1C–2C).

**Figure 6 foods-13-03566-f006:**
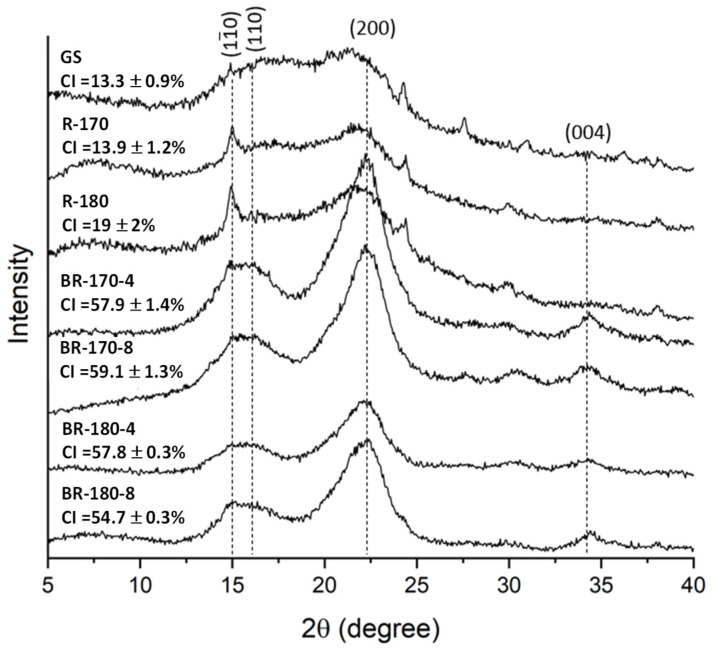
XRD spectra and crystallinity index (CI) of the grape stalk (GS), non-bleached samples (R-170 and R-180) and the two-cycles-bleached samples (BR) under different conditions (4 or 8% *v*/*v* alkaline hydrogen peroxide solutions).

**Figure 7 foods-13-03566-f007:**
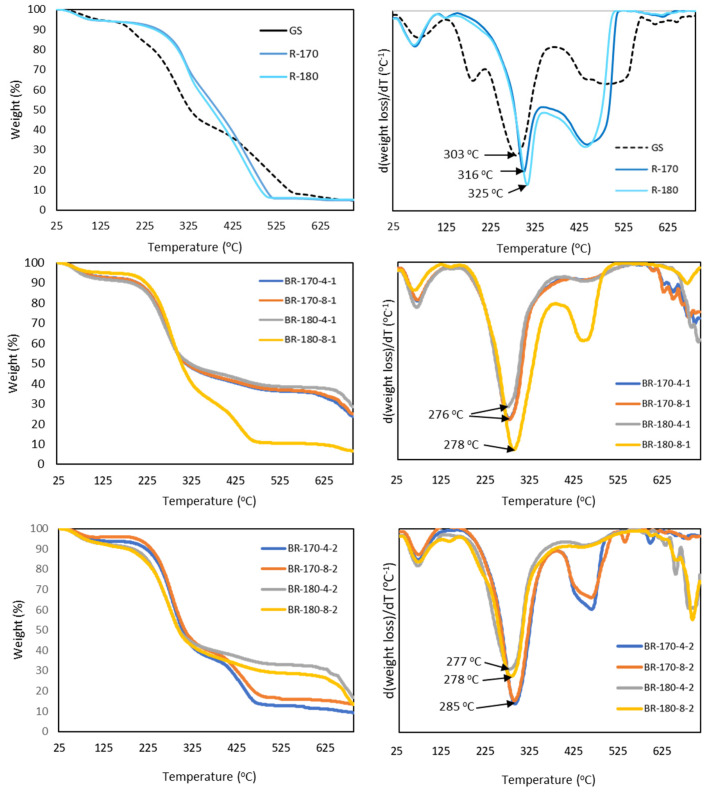
Thermogravimetric analysis (TGA) and DGTA curves of the grape stalk (GS), non-bleached subcritical water extraction residues (R-170 and R-180) and the bleached residues (BR) after one and two bleaching cycles with 4% or 8% (*v*/*v*) alkaline hydrogen peroxide solutions. Bleached residues are coded as BR temperature of subcritical water extraction concentration (%) of hydrogen peroxide-number of 1 h bleaching cycles.

**Table 1 foods-13-03566-t001:** Extractives, structural components, and ashes of winemaking grape stalks (GS) and the subcritical water extraction insoluble residues obtained at 170 °C (R-170) and 180 °C (R-180). Mean values ± standard deviation.

Sample	Extractives in Water (g/100 g Sample)	Cellulose (g/100 g Sample)	Hemicellulose (g/100 g Sample)	Lignin (g/100 g Sample)	Ashes (g/100 g Sample)
GS	52.2 ± 0.0 ^a^	22 ± 4 ^a^	7 ± 4 ^a^	14.7 ± 1.3 ^a^	7.6 ± 0.1 ^a^
R-170	25.9 ± 0.0 ^b^	32 ± 4 ^b^	1.6 ± 0.1 ^b^	41.3 ± 1.6 ^b^	2.9 ± 0.1 ^b^
R-180	21.2 ± 0.1 ^b^	24.2 ± 1.8 ^b^	0.4 ± 0.1 ^b^	40 ± 0.2 ^b^	2.9 ± 0.1 ^b^

Different letters (a, b) in the same column indicate significant differences among samples (*p* < 0.05).

**Table 2 foods-13-03566-t002:** Total phenolic content (TPC) and EC50 of the grape stalk (GS) and the subcritical water extraction fractions obtained at 170 °C and 180 °C: Insoluble residues (R-170, R-180) and extracts (E-170, E-180). Mean values ± standard deviation.

Sample	TPC (g GAE/100 g Sample Solids)	TPC (g GAE/100 g GS)	EC 50 (mg Sample Solids/mg DPPH)	EC 50 (mg GS/mg DPPH)
GS	9.3 ± 0.6 ^4^	3.8 ± 0.2 ^1^	0.64 ± 0.07 ^1^	27 ± 3 ^1^
R-170	17.8 ± 0.3 ^b,2^	1.3 ± 0.0 ^b,5^	0.38 ± 0.05 ^a,2^	2.8 ± 0.4 ^a,4^
R-180	19.7 ± 0.3 ^a,1^	1.6 ± 0.0 ^a,4^	0.38 ± 0.01 ^a,2,3^	3.11 ± 0.07 ^a,4^
E-170	8.3 ± 0.0 ^b,5^	2.7 ± 0.0 ^b,3^	0.42 ± 0.00 ^a,2,3^	13.33 ± 0.13 ^a,2^
E-180	11.1 ± 0.1 ^a,3^	3.2 ± 0.0 ^a,2^	0.30 ± 0.01 ^b,2^	8.9 ± 0.4 ^b,3^

Different superscripts indicate significant differences among all samples in each column (1–5) or due to temperature (a, b) for the same subcritical water extraction fraction (*p* < 0.05).

**Table 3 foods-13-03566-t003:** Composition of grape stalk (GS), subcritical water extraction residues (SWE) at 170 °C (R-170) and 180 °C (R-180), and the corresponding bleached residues (BR), as referred to the initial GS basis, together with the global mas yield of each fraction. Mean values ± standard deviation. Bleached residues are coded as the BR temperature of SWE concentration (% *v*/*v*) of alkaline hydrogen peroxide number of 1 h bleaching cycles.

Sample	Cellulose (g/100 g GS)	Klason Lignin (g/100 g GS)	Ash (g/100 g GS)	Global Yield (g Fraction/100g GS)
GS	22.3 ± 4 ^e^	14.4 ± 1.3 ^e^	7.6 ± 0.1 ^e^	-
R-170	17.1 ± 1.9 ^bcd^	11.0 ± 0.9 ^abc^	0.8 ± 0.0 ^abc^	54 ± 2 ^d^
BR-170-4-1	18.9 ± 0.3 ^bcd^	6.0 ± 0.0 ^abc^	0.5 ± 0.0 ^abc^	30 ± 2 ^c^
BR-170-4-2	15.0 ± 2.5 ^bcd^	4.4 ± 0.5 ^abc^	0.3 ± 0.1 ^abc^	21 ± 3 ^b^
BR-170-8-1	14.3 ± 1.1 ^bcd^	7.3 ± 0.8 ^d^	0.6 ± 0.0 ^d^	25 ± 2 ^b^
BR-170-8-2	13.0 ± 1.5 ^bcd^	5.4 ± 0.5 ^abc^	0.3 ± 0.0 ^abc^	22 ± 2 ^b^
R-180	14 ± 1.0 ^bcd^	12 ± 0.1 ^abc^	0.8 ± 0.1 ^abc^	58 ± 2 ^d^
BR-180-4-1	12.4 ± 1.4 ^bcd^	6.8 ± 0.1 ^abc^	0.3 ± 0.0 ^abc^	23 ± 2 ^b^
BR-180-4-2	9.2 ± 1.1 ^ab^	3.5 ± 0.1 ^ab^	0.2 ± 0.0 ^ab^	15 ± 2 ^a^
BR-180-8-1	12 ± 2.0 ^bcd^	5.7 ± 0.7 ^abc^	0.3 ± 0.0 ^abc^	22 ± 2 ^b^
BR-180-8-2	10.3 ± 2.0 ^cd^	3.1 ± 0.5 ^cd^	0.3 ± 0.1 ^cd^	16 ± 2 ^a^

^a–e^ Different letters in the same column indicate significant differences among samples (*p* < 0.05).

## Data Availability

The original contributions presented in the study are included in the article, further inquiries can be directed to the corresponding author.

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
