# Peer review of "Fractionation of Winemaking Grape Stalks by Subcritical Water Extraction to Obtain Added-Value Products"

_foods, 2024, doi:10.3390/foods13223566_

Round 1

Reviewer 1 Report

Comments and Suggestions for Authors

In the current study, the authors investigated the feasibility of using subcritical water extraction to isolated bioactives such as polyphenolic compounds, cellulosic fibres, etc; and tested their anti-oxidant activities or their stuctures. Overall, this study is very straight-forward with some interesting findings. However, the authors should address the following two key questions:

1) There is no comparation between subcritical water extraction with hot water extraction or solvent extraction at all. Therefore, the clamis of advantages of subcritical water extraction could not be supported. 

2) Any infulence of high temperature during the extraction on the polyphenolic compounds? since normally these polyphenolic compounds should be unstable at such high temperature. 

Author Response

Comment 1: There is no comparison between subcritical water extraction with hot water extraction or solvent extraction at all. Therefore, the claims of advantages of subcritical water extraction could not be supported.

Response 1: In the text, phenolic contents of grape stalks extracted by other methods are given. These values were less than 0.4 g GAE per 100g of grape stalk, which is lower than that obtained in the present study by SWE. This has been included in the revised manuscript (lines 346-349). 

Comment 2: Any influence of high temperature during the extraction on the polyphenolic compounds? since normally these polyphenolic compounds should be unstable at such high temperature.

Response 2: Temperature has the most significant effect on SWE efficiency, and thus, it has been optimized for different plant matrices. This has been referenced in the revised manuscript (lines 359-361). 

Reviewer 2 Report

Comments and Suggestions for Authors

Dear Authors,

The selected topic on valorization of grapewine stalks is not novel but you have proposed interesting combination of subcritical water extraction and bleaching process in order to achieve maximal recovery of cellulose, which should be mentioned in title or at least in keywords. Please find technical comments and suggestions in attached file.

Kind regards

Author Response

These are the responses to the comments included in the attached pdf file:

f1. If the affiliation institution is the same, there is no need to repeat it 4 times.

Response: It has been corrected in the revised manuscript.

f2. Usually one researcher is correspondence.

Response: It has been corrected in the revised manuscript.

f3. Font issues.

Response: We do not understand the comment. Could you please clarify?

f4. solid/liquid ratio 1:10.

Response: It has been changed to solids/water mass ratio 1:10 to specify that is a mass ratio (not volume).

f5. 6

Response: It has been changed in the revised manuscript.

f6 & f7. S/L ratio

Response: It has been changed to S/L mass ratio to specify that it is a mass ratio (not volume).

f8. This procedure has already been mentioned.

Response: It has not been already mentioned. The previous method is for extractives (not for structural carbohydrates).

f9. Soxhlet extraction is used for recovery of lipid fraction.

Response: Soxhlet technique is an extraction procedure that is also used to recover water extractives following the NREL standard (NREL/TP-510-42619-2008).

f10. Italic.

Response: It has been corrected.

f11. This sentence should not be the first in this chapter, it should stand at the end of this paragraph.

Response: The paragraph has been rearranged for more clarity.

f12. This should be in the Methods section.

Response: This was indicated in the methods section, and it is mentioned again in this section to help discuss the reported results.

f13. The reference is incomplete.

Response: It has been completed.

f14. Space

Response: It has been corrected,

f15. Please exclude abbreviations.

Response: They have been excluded.

f15. Exclude if possible.

Response: It has been excluded.

Reviewer 3 Report

Comments and Suggestions for Authors

The manuscript deals with the Fractionation of winemaking grape stalks by subcritical water extraction to obtain added-value products. This study explores the potential of using combined treatments of subcritical water extraction and hydrogen peroxide delignification cycles to fractionate winemaking grape stalks, producing phenol-rich aqueous extracts and cellulosic fractions, both valuable in developing food packing materials. Both extracts and solid residues were characterised in terms of their phenolic richness and antioxidant activity. The structural components (cellulose, hemicellulose, and lignin) were analysed in the solid residues of the extraction and at the different steps of the bleaching process.

The topic is interesting and catches the audience’s attention, but it needs to improve according to the suggested comments.

Abstract:

Lines 20 to 24: The first bleaching cycle led to a significant reduction in the lignin content at both SWE temperatures. The cellulose purity was higher in samples obtained at 170 ºC and bleached with 4% alkaline hydrogen peroxide than in those submitted to SWE at 180 ºC, when greater cellulose oxidation during the bleaching step seems to be promoted regardless of the oxidant concentration. Reconcile these sentences with a proper conclusion.

Introduction:

Lines 29 to 30: Grape is one of the most abundant fruit crops in the world. In 2019, the world production of this fruit was 77.8 million tons, of which 57% were wine grapes. Add the latest production details, if possible, after a few sentences.

The lacking part of the introduction is a clear statement of the research gap or need for the current study. While the introduction mentions several studies but does not explicitly state the specific gap or problem the current study aims to address. In addition, there is no direct link between the previous studies and the objectives of the current study.

Add recent references.

Lines 111 to 117: This study explores the potential of using combined treatments of subcritical water extraction and hydrogen peroxide delignification cycles to fractionate winemaking grape stalks, producing phenol-rich aqueous extracts and cellulosic fractions, both valuable in developing food packing materials. Both extracts and solid residues were characterised in terms of their phenolic richness and antioxidant activity. The structural components (cellulose, hemicellulose, and lignin) were analysed in the solid residues of the extraction and at the different steps of the bleaching process. Rewrite this paragraph with proper concluding points and some future directions.

Conclusion:

Line 363 to 364: Recovery of cellulose from the insoluble SWE fractions, obtained at 170 or 180 °C, was also possible by applying a greener bleaching/delignification process with alkaline hydrogen peroxide. After two bleaching cycles, no notable delignification progress was observed in the samples, as the bleaching yield or whiteness index did not significantly change in the further cycles. The first bleaching cycle led to a significant reduction (65-75 %) in the lignin content in the SWE residues obtained at both temperatures. Please clarify and be brief.

Comments on the Quality of English Language

Minor English editing is needed.

Author Response

Comment 1:

Abstract: Lines 20 to 24: The first bleaching cycle led to a significant reduction in the lignin content at both SWE temperatures. The cellulose purity was higher in samples obtained at 170 ºC and bleached with 4% alkaline hydrogen peroxide than in those submitted to SWE at 180 ºC, when greater cellulose oxidation during the bleaching step seems to be promoted regardless of the oxidant concentration. Reconcile these sentences with a proper conclusion.

Response 1: This part has been rewritten following the reviewer’s suggestion (lines 16-20).

Comment 2: Introduction:

Lines 29 to 30: Grape is one of the most abundant fruit crops in the world. In 2019, the world production of this fruit was 77.8 million tons, of which 57% were wine grapes. Add the latest production details, if possible, after a few sentences.

Response 2: The latest available production details have been included with the corresponding new reference (lines 25-28). As a consequence, the references have been renumbered.

Comment 3: The lacking part of the introduction is a clear statement of the research gap or need for the current study. While the introduction mentions several studies but does not explicitly state the specific gap or problem the current study aims to address. In addition, there is no direct link between the previous studies and the objectives of the current study.

Response 3: This part has been rewritten (lines 97; 108-111).

Comment 4: Add recent references.

Response 4: The most recent references as regards the revalorization of grape stalks and the use of SWE and bleaching were included in the manuscript.

Comment 5:

Lines 111 to 117: This study explores the potential of using combined treatments of subcritical water extraction and hydrogen peroxide delignification cycles to fractionate winemaking grape stalks, producing phenol-rich aqueous extracts and cellulosic fractions, both valuable in developing food packing materials. Both extracts and solid residues were characterised in terms of their phenolic richness and antioxidant activity. The structural components (cellulose, hemicellulose, and lignin) were analysed in the solid residues of the extraction and at the different steps of the bleaching process. Rewrite this paragraph with proper concluding points and some future directions.

Response 5: This paragraph has been rewritten (lines 111-118).

Comment 6: Conclusion: Line 363 to 364: Recovery of cellulose from the insoluble SWE fractions, obtained at 170 or 180 °C, was also possible by applying a greener bleaching/delignification process with alkaline hydrogen peroxide. After two bleaching cycles, no notable delignification progress was observed in the samples, as the bleaching yield or whiteness index did not significantly change in the further cycles. The first bleaching cycle led to a significant reduction (65-75 %) in the lignin content in the SWE residues obtained at both temperatures. Please clarify and be brief.

Response 6: This paragraph has been rewritten following the reviewer’s suggestion.

Reviewer 4 Report

Comments and Suggestions for Authors

This manuscript was logically conceived, and the results looks reasonable.

1. Please, check the format of tables and figures according to journal requirements. And, the abbreviations used in the tables and figures should be explained in the footnote of tables and/or figure captions to make them standalone.

2. The results of standard deviation and significance analysis of the data should be provided to better support the conclusions, including data in Figure 1 (extraction yield), Figure 2a, Figure 3ab, Table 3 and Figure 6  (crystallinity index (CI)).

3. Figure 5 should be given as clear picture.

4. The number of significant digits in the result should be consistent.

5. Revise carefully reference style. For example, volume (number included or not), Journal name (italic or not), etc.

Author Response

Comment 1: Please, check the format of tables and figures according to journal requirements. And, the abbreviations used in the tables and figures should be explained in the footnote of tables and/or figure captions to make them standalone.

Response 1: The format of all tables and figures has been checked, and the abbreviations have been explained.

Comment 2: The results of standard deviation and significance analysis of the data should be provided to better support the conclusions, including data in Figure 1 (extraction yield), Figure 2a, Figure 3ab, Table 3 and Figure 6 (crystallinity index (CI)).

Response 2: The standard deviation values have been included in Figure 1 and Figure 6. The values of the LSD intervals in figure 2a and figure 3ab have been included for each one of the analysed variables in the figure caption. The significance analyses in Table 3 have been explained as a footnote.

Comment 3: Figure 5 should be given as clear picture.

Response 3: Thank you for the comment. From our point of view, the Figure is clear enough to discuss the FTIR results.

Comment 4: The number of significant digits in the result should be consistent.

Response 4: The number of significant digits in the average reported results for each parameter is consistent with the significant figures of the corresponding standard deviation values.

Comment 5: Revise carefully the reference style. For example, volume (number included or not), Journal name (italic or not), etc.

Response 5: The references have been carefully revised.

Round 2

Reviewer 1 Report

Comments and Suggestions for Authors

The authors have addressed my comments and I think it is suitable to be published.

Author Response

Comment 1: The authors have addressed my comments and I think it is suitable to be published.

Response: Thank you for your comment. 

Reviewer 4 Report

Comments and Suggestions for Authors

1. The results of standard deviation and significance analysis of the data should be provided to better support the conclusions, including data in Figure 2a, and Table 3 (Global Yield).

Author Response

Comment 1. The results of standard deviation and significance analysis of the data should be provided to better support the conclusions, including data in Figure 2a, and Table 3 (Global Yield).

Response: The results of the standard deviation and significance analysis of Global Yield have been provided in Table 3. The results of the significance analysis of the data shown in Figure 2a have been provided as LSD intervals in the figure captions.